# The SARS-CoV-2 main protease doesn't induce cell death in human cells *in vitro*

**Alexey Komissarov**⬤, **Maria Karaseva, Marina Roschina, Sergey Kostrov, Ilya Demidyuk***

Institute of Molecular Genetics of National Research Centre Kurchatov Institute, Moscow, Russian Federation

* duk@img.ras.ru

## Abstract

Severe acute respiratory syndrome coronavirus 2 (SARS-CoV-2) is the cause of coronavirus disease 2019 (COVID-19) which has extremely rapidly spread worldwide. In order to develop the effective antiviral therapies, it is required to understand the molecular mechanisms of the SARS-CoV-2 pathogenesis. The main protease, or 3C-like protease (3CL$^{pro}$), plays the essential role in the coronavirus replication that makes the enzyme a promising therapeutic target. Viral enzymes are known to be multifunctional. Particularly, 3CL$^{pro}$ of SARS-CoV was shown to induce apoptosis in addition to its main function. In the present study we analyzed the cytotoxicity of active SARS-CoV-2 3CL$^{pro}$ and its inactivated form upon their individual expression in four human cell lines. For this purpose, we constructed a protein biosensor which allows to detect the proteolytic activity of SARS-CoV-2 3CL$^{pro}$ and confirmed the expression of the active protease in all cell lines used. We studied viability and morphology of the cells and found that both active and inactivated enzyme variants induce no cell death in contrast to the homologous 3CL protease of SARS-CoV. These results indicate that SARS-CoV-2 3CL$^{pro}$ is unlikely contribute to the cytopathic effect observed during viral infection directly.

## Introduction

Severe acute respiratory syndrome coronavirus 2 (SARS-CoV-2) belongs to the *Betacoronavirus* genus and is the cause of coronavirus disease 2019 (COVID-19) which has extremely rapidly spread worldwide since December 2019. Due to continuously increasing number of infected and lethal cases there is an urgent need to develop the effective antiviral drugs. For this purpose, it is required to understand the mechanisms of the SARS-CoV-2 pathogenesis, particularly the involvement and role of the individual virus-derived molecules. The SARS-CoV-2 main protease, or 3C-like protease (3CL$^{pro}$), is a cysteine protease that plays the essential role in virus replication through hydrolyses of the viral polyproteins into the mature functional proteins. This together with the high homology of the 3CL$^{pro}$ between coronaviruses [1] makes the enzyme a prospective therapeutic target for the treatment of the coronavirus infection [2,3].

Viral enzymes are known to be multifunctional and often possess additional functions besides the major one. A prime example is the picornaviral 3C proteases [4]–the homologs of

**Funding:** This work was supported in part by the Kurchatov Genomic Center Development Program

of the Ministry of Science and Higher Education of the Russian Federation (https://minobrnauki.gov. ru/action/fntp/) through a grant awarded to SK (agreement no. 075-15-2019-1664). No additional external funding was received for this study. The funders had no role in study design, data collection and analysis, decision to publish, or preparation of the manuscript.

**Competing interests:** The authors have declared that no competing interests exist.

betacoronaviral 3CL^pro; it is from this homology that the name of the 3CL^pro originates. In addition to the main function 3C proteases of various picornaviruses are able to induce cell death in human cells [5–10]. The 3CL^pro from SARS-CoV, another member of betacoronaviruses that caused the pneumonia epidemic in 2002–2003, was also shown to induce caspase-dependent apoptosis [11]. Since there is a 96% sequence identity between SARS-CoV 3CL^pro and SARS-CoV-2 3CL^pro [12], the latter enzyme potentially may also be cytotoxic. Despite 3CL^pro is currently actively studying, it is still unclear whether the enzyme possesses cytotoxic effects. In several studies no cytotoxicity was reported upon individual expression of 3CL^pro [13,14], while in one preprint cytotoxic effects of the enzyme were observed [15]. However, since these studies were focused on the examination of the 3CL^pro inhibitors, cell morphology and observed cytotoxic effect were not characterized. In the present study we expressed SARS-CoV-2 3CL^pro and its inactivated mutant variant in human cells and characterized cell morphology and viability. We found that the active enzyme doesn't induce cell death or alter cell morphology.

## Materials and methods

### Plasmids

A DNA fragment containing *EcoR*I restriction site, the Kozak sequence (GCCACC), start codon (ATG), SARS-CoV-2 3C-like protease (3CL^pro) sequence (GenBank MN908947 genome sequence, nucleotides 10055–10972), stop codon (TAA), and *Kpn*I restriction site was synthesized by Evrogen (Russia). In addition, similar DNA fragment was generated bearing the mutant 3CL^pro sequence (*m3CL^pro*) encoding inactivated 3CL^pro with the Cys145-Ala substitution (mature enzyme numbering). The fragments were cloned into pCI expression vector (Promega, USA) to the *Eco*RI-*Kpn*I sites. The resulting plasmids expressing 3CL^pro and m3CL^pro were referred to as p3CL and pm3CL, respectively.

The plasmid p3C for the expression of human hepatitis A virus 3C protease (3Cpro) was generated as follows. A DNA fragment containing *Eco*RI restriction site, the Kozak sequence (GCCACC), 3Cpro gene, and *Kpn*I restriction site was produced using PCR with oligonucleotides `GACTGAATTCGCCACCATGTCAACTCTAGAAATAGCAGG` and `CAACGGTACCTTACTGACTTTCAATTTTCTTATCAATG` (Evrogen, Russia) as primers, and pBI-EGFP-3C plasmid [10] as a matrix. The fragment was cloned to the pCI vector to the *Eco*RI-*Kpn*I sites. The plasmid p3Cmut was constructed in the same way except that pBI-EGFP-3Cmut [10] was used as a source of the 3Cmut gene encoding inactivated 3Cpro with the Cys172-Ala substitution. The plasmid pCI-EGFP was constructed previously [16].

The plasmid pGlo-3CL was constructed based on the pGloSensor-30F-DEVDG vector (Promega, USA). Briefly, oligonucleotides `GATCCGCCGTGCTGCAGTCA` and `AGCTTGACTGCAGCACGGCG` (Evrogen, Russia) were mixed in equimolar ratio (final concentration 10 μM each), the mixture was heated at 95 ˚C followed by gradual cooling down to room temperature. The resulting DNA duplex was cloned into pGloSensor-30F-DEVDG vector predigested with *Bam*HI and *Hind*III enzymes.

The structure of all generated plasmids was confirmed by sequencing using ABI PRISM BigDye Terminator v. 3.1 reagents on a 3730 DNA Analyzer (Applied Biosystems, USA). The plasmids were amplified in *E. coli* TG1 cells and purified using a Plasmid Miniprep kit (Evrogen, Russia).

### Cell cultures and transfection

Human embryonic kidney HEK293, human cervical cancer HeLa (M-HeLa clone 11), and human adenocarcinomic alveolar basal epithelial A549 cell lines were obtained from the

Russian Cell Culture Collection (St. Petersburg, Russia). Non-small-cell lung cancer cell line Calu-1 was kindly provided by Dr. Evgeniy Kopantzev (Shemyakin-Ovchinnikov Institute of Bioorganic Chemistry of the Russian Academy of Sciences). The cells were cultured in DMEM/F-12 medium supplemented with 10% fetal bovine serum and 0.3 mg/ml glutamine at 37 ˚C in humidified atmosphere with 5% $CO_2$. For transfection, the cells were cultured in 96- or 6-well plates (Corning, USA) for 20–24 h until 70–90% confluence. For transfection plasmid DNA-Lipofectamine 2000 (ThermoFisher Scientific, USA) complexes were prepared following the manufacturer's protocol in serum-free OptiMEM medium (ThermoFisher Scientific, USA) and then added into the wells; 4 h after the addition of the complexes the medium was replaced with the fresh one.

## RNA isolation, cDNA synthesis and quantitative PCR

Cells were transfected as described above in 6-well plates, 24 h post transfection (p.t.) were detached by incubation with 1 mL of phosphate buffered saline (PBS) supplemented with 0.2 g/L ethylenediaminetetraacetic acid (EDTA) for 10 min at 37 ˚C in humidified atmosphere with 5% $CO_2$, then were pelleted by centrifugation at 400g for 5 min. Total RNA was isolated from the cell pellets using RNeasy Mini Kit (Qiagen, USA) according to the manufacturer's standard protocol. Isolated total RNA was eluted with nuclease free water and RNA concentration was determined by absorbance at 260 nm using the Agilent 8453 UV-Vis spectrophotometer (Agilent Technologies, USA). For the absorbance to concentration conversion the equation was used: 1.0 A260 = 40 μg/mL of total RNA. To remove genomic DNA, 2 μg of total RNA were treated with 20 units of bovine DNase I (Sigma, Germany) in 20 μL of reaction buffer (10 mM Tris-HCl, pH 7.5, 1 mM $MgCl_2$, 1 mM $CaCl_2$) for 20 min at room temperature, then DNase I was inactivated by incubation at 65 ˚C for 10 min. We checked the completeness of DNA removal in each sample using PCR in duplicate with 100 ng of total RNA prior to reverse transcription. The resulting total RNA was reverse transcribed using REVERTA-L kit (AmpliSens, Russia) provided with hexanucleotides of random sequence according to the manufacturer's protocol.

The quantitative PCR was performed on a CFX96 Touch real-time PCR detection system (Bio-Rad, USA) using primers and probes listed in Table 1. The 25 μL reaction mixture contained approximately 100 ng of cDNA, 10 pmol of each primer and probe, deoxynucleotide triphosphates (2.5 mM each), 2.5 mM $MgCl_2$, 2.5 U of SynTaq DNA-polymerase, 2.5 μL of 10X SynTaq DNA-polymerase reaction buffer with SYBR Green (Syntol, Russia). The PCR program was performed as follows: 5 min at 95˚C; then, 42 cycles, each comprising 15 s at 95 ˚C, 50 s at 60 ˚C. To check the specificity of amplification the melting curve was generated following the last cycle by heating from 65 to 95 ˚C in increments of 0.5 ˚C (S1A and S1B Fig). Each

**Table 1. Oligonucleotides used for quantitative real-time PCR.**

| Target | | Sequence |
|---|---|---|
| *3CL^pro / m3CL^pro* | Forward primer: | TTGGGTCGCAGTTCTTGTTTG |
| | Reverse primer: | TGCCTTGACATTCTCGATGGT |
| | Probe: | VIC-TCGCTGTGATCGTCACTTGACAATG-BHQ2 |
| *Ubiquitin C (UBC)* | Forward primer: | CATGCTGGCACAGACTTAGAAG |
| | Reverse primer: | GCAGCGTACAACCAAGCTAAAC |
| | Probe: | VIC-ACAGGCAAACAGCACAAGCAGC-BHQ2 |

All oligonucleotides were synthesized by DNK-sintez (Russia). VIC– 2′-chloro-7′phenyl-1,4-dichloro-6-carboxy-fluorescein; ROX—carboxyrhodamine; BHQ2 –Black Hole Quencher 2.

sample was analyzed in triplicates. Results were processed using the CFX-Manager software (Bio-Rad, USA). The obtained Ct values for SARS-CoV-2 3CL$^{pro}$ and m3CL$^{rpo}$ genes were normalized against Ct values for the ubiquitin C (UBC) reference gene using standard delta-delta Ct method.

## Assay of the 3CL activity

Cells were co-transfected with pGlo-3CL and p3CL, pm3CL or intact pCI plasmid as described above in 96-well white plates with clear bottom (Greiner Bio-One, Germany). Next, 24 h p.t. luciferase activity was analyzed in transfected cultures using a GloSensor reagent (Promega, USA). Briefly, the medium in the wells of 96-well plates was replaced with 100 μl of growth medium supplemented with 2% (v/v) GloSensor reagent and the luminescence was recorded using an Infinite M200 Pro microplate reader (Tecan, Switzerland) within 20 min of incubation at 37 ˚C in humidified atmosphere with 5% $CO_2$.

## Cell viability assay

Cells were transfected as described above in 96-well plates, then 24 and 48 h p.t. the viability of the cells in transfected and non-transfected cultures was determined using a CellTiter 96 AQueous One Solution Cell Proliferation Assay kit (Promega, USA). Briefly, the medium in the wells of 96-well plates was replaced with 100 μl of PBS, and 20 μl aliquots of the CellTiter 96 reagent were added per well. The absorbance was recorded at 490 nm with the Multiskan Ascent plate reader (ThermoFisher Scientific, USA) immediately after the CellTiter 96 reagent addition and after 1 h of incubation at 37 ˚C in humidified atmosphere with 5% $CO_2$.

## Flow cytometry

The medium in wells of 96-well plate 24 or 48 h p.t. was replaced with 100 μL of PBS supplemented with 0.2 g/L EDTA and 50 μM 1,1′,3,3,3′,3′-hexamethylindodicarbocyanine iodide dye (Mito, ThermoFisher Scientific, USA), and the plate was incubated at 37 ˚C in humidified atmosphere with 5% $CO_2$ for 20 min. After the detachment, the cells were transferred to 0.5 mL tubes, stained with 1 μM propidium iodide (PI, Sigma, Germany) for 5 min, and then analyzed using Accuri C6 flow cytometer (Becton Dickinson, USA). For each sample at least 20 000 events corresponding to single cells by forward and side light scatter were acquired. The Mito fluorescence was detected using 640-nm excitation and 660–685 nm emission filter, for PI detection 488-nm excitation and detection at 600–620 nm were used. Raw data were analyzed using Accuri C6 Software (Becton Dickinson, USA) and FlowJo (FlowJo LLC, USA). For apoptosis induction the cells were incubated with 150 ng/mL tumor necrosis factor α (TNFα, Sigma, Germany) in combination with 2 μg/mL cycloheximide (CHX, Sigma, Germany) or 1 μM staurosporine (Sigma, Germany) for 24 h prior to flow cytometry analysis.

## Statistical analysis

Comparison between unrelated groups was carried out using nonparametric two-tailed Mann-Whitney U test. Differences were considered significant if the p-value was less than 0.05. Statistical analysis was performed using R (R Core Team (2020). R: A language and environment for statistical computing. R Foundation for Statistical Computing, Vienna, Austria; https://www.R-project.org/) and RStudio version 1.3.1093 (RStudio Team (2019). RStudio: Integrated Development for R. RStudio, Inc., Boston, MA; http://www.rstudio.com/) software.

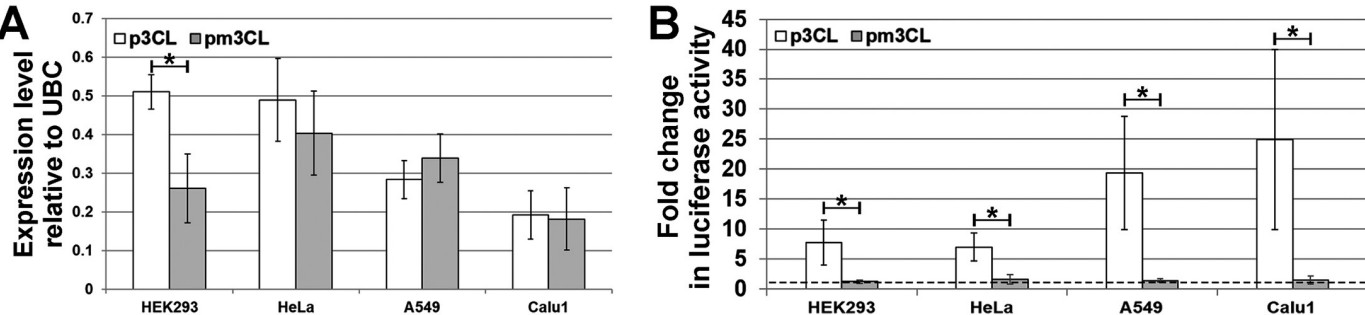

**Fig 1. Analysis of the 3CLpro and m3CLpro mRNA expression and activity.** HEK293, HeLa, A549, and Calu1 cells were transfected with p3CL or pm3CL plasmids. Total RNA was isolated 24 h post transfection and expression of target genes was detected using quantitative PCR with reverse transcription (A). Expression of 3CLpro and m3CLpro was normalized against ubiquitin C (UBC) reference gene and analyzed for each cell line used. Values are represented as mean ± SD of three independent measurements. The proteolytic activity of 3CL$^{pro}$ was assayed using GloSensor technology (B). Cells were co-transfected with pGlo-3CL and p3CL or pGlo-3CL and pm3CL constructs and 24 h p.t. luciferase activity in transfected cells was analyzed. Fold changes in luciferase activity relative to control cells, transfected with pGlo-3CL and pCI, after 20 min of incubation with GloSensor reagent are shown. Values are represented as mean ± SD of two independent experiments with triplicates (n = 6).

## Results and discussion

We constructed the p3CL plasmid for the expression of the 3CL$^{pro}$ under the constitutive cytomegalovirus promoter. The analogous pm3CL plasmid that provides for the expression of the mutant 3CL$^{pro}$ (m3CL$^{pro}$) inactivated due to Cys145 to Ala substitution in the catalytic site was created as a control. The X-ray crystal structure of the enzyme determined that the Cys145 is a key residue in the catalytic site [17] and analogous substitution in 3CL$^{pro}$ from SARS-CoV was shown to cause the loss of the proteolytic activity [18]. The plasmids were introduced into two the most commonly used (HEK293 and HeLa) and two lung-derived (A549 and Calu1) human cell lines and expression of the corresponding genes was confirmed at mRNA level using quantitative PCR (Fig 1A and S1 Fig). We showed that there was no statistically significant difference in the mRNA expression levels between 3CL$^{pro}$ and m3CL$^{pro}$ in all cell lines except for HEK293. Additionally, expression levels between several cell lines differed and there was a tendency for the levels to be higher in HEK293 and HeLa cells then in A549 and Calu1. However, these differences are minimal and unlikely reflect features of the 3CL$^{pro}$ and host-cell interactions, and seem to originate from other reasons, e.g., due to different transfection efficiencies of the cell lines and transfection agents used.

We also analyzed the proteolytic activity of 3CL$^{pro}$ in transfected cells. For this purpose, we created a luciferase-based biosensor similar to the one described for 3C-like protease of the Middle East respiratory syndrome coronavirus [19]. This biosensor is based on the GloSensor technology developed by Promega and encoded by pGlo-3CL plasmid. Within the human cells *in vitro* the plasmid provides for the expression of the inactive circular permutated firefly luciferase [20]. The wild-type N- and C-termini of the enzyme are linked by a polypeptide containing the 3CL$^{pro}$ recognition site (AVLQ↓S) corresponding to the SARS-CoV-2 polyprotein nsp4/5 junction. Thus, a conformational change in the biosensor upon cleavage by 3CL$^{pro}$ restores the luciferase catalytic activity. A biosensor utilizing the same principle was recently created independently by another group [21]. We showed that robust luciferase signal was observed in cells co-expressing the biosensor and 3CL$^{pro}$, while no signal was detected in cells both expressing inactive m3CL$^{pro}$ and transfected with intact pCI vector (Fig 1B and S2 Fig). Noteworthy, that luciferase activity in lung-derived cell lines A549 and Calu1 was considerably higher than in other cell lines. However, our data are insufficient to confirm the relationship between the cell type and 3CL$^{pro}$ activity, and this issue should be studied in more details.

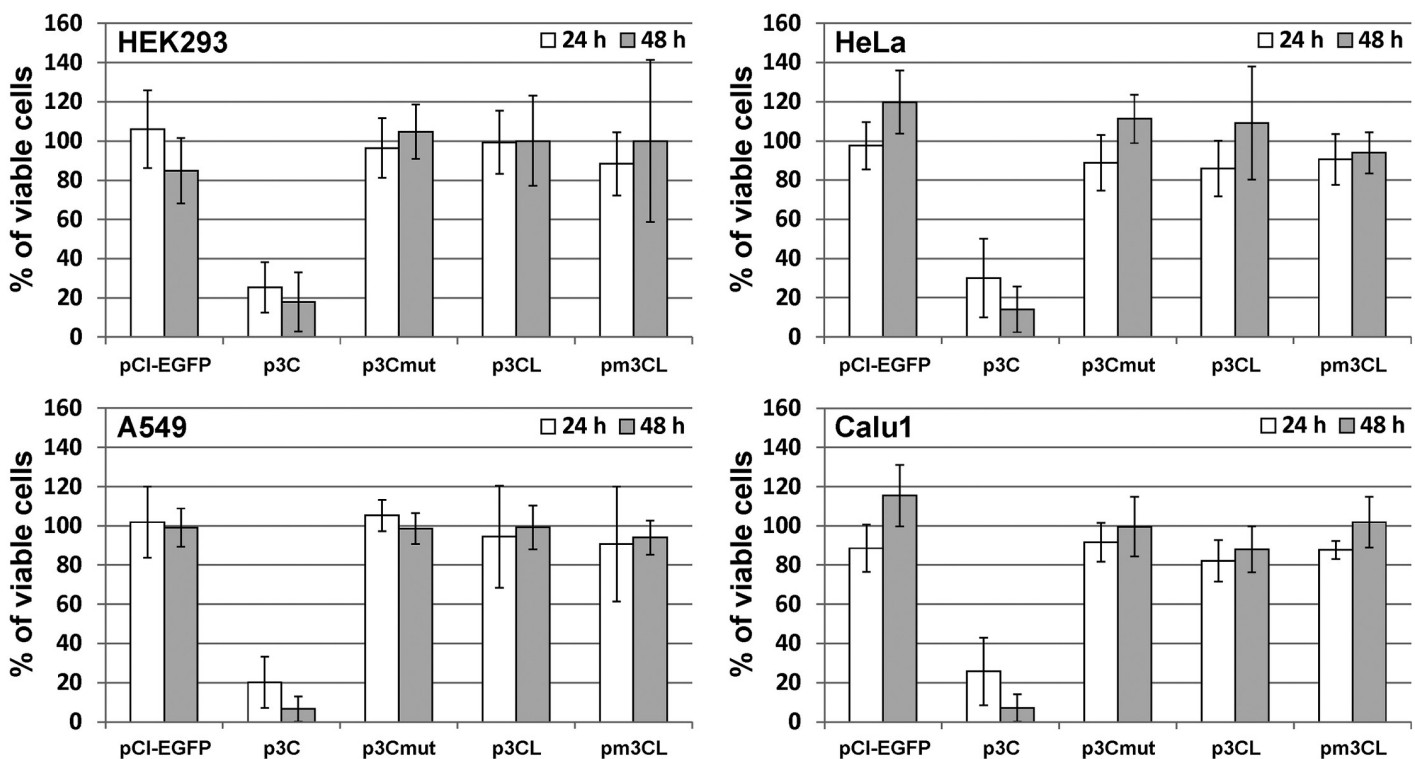

**Fig 2. Viability estimation in transfected cultures.** HEK293, HeLa, A549 and Calu1 cells were transfected with the indicated plasmids and 24 h and 48 h post transfection metabolic activity was analyzed using CellTiter 96 reagent. Results are expressed as the percentage of viable cells relative to non-transfected cultures. Values are represented as mean ± SD of two independent experiments with triplicates (n = 6).

The cytotoxic effect of the 3CL$^{pro}$ was evaluated 24 and 48 h post transfection (p.t.) with p3CL and pm3CL plasmids (Fig 2). Additionally, the cells were transfected with p3C, p3Cmut, and pCI-EGFP plasmids. The p3C plasmid which provides the expression of human hepatitis A virus 3C protease (3Cpro), the 3CL$^{pro}$ homolog, served as a positive control in the experiment since we have shown previously in similar experimental system that this enzyme exerts a strong cytotoxic effect [22]; p3Cmut and pCI-EGFP plasmids providing the expression of inactivated mutant 3Cpro (3Cmut) and enhanced green fluorescent protein (EGFP), respectively, served as a negative control since these proteins were shown to possess no cytotoxicity [16,22]. For all cell lines used the proportion of viable cells in cultures transfected with p3C compared to non-transfected cells was approximately 30% at 24 h and 15% at 48 h p.t., thus demonstrating significant cytotoxic effect, whereas no cytotoxicity was observed both 24 h and 48 h p.t. in case of the other plasmids, including p3CL. However, taking into account that efficiency of the transient transfection never achieves 100% [23], the potential cytotoxic effect of 3CL$^{pro}$ can be masked by the proliferation of non-transfected cells in the experimental system used. For this reason, we further estimated the viability of the transfected cells only.

In order to visualize the transfected cells in the cultures the p3CL, pm3CL, and p3Cmut plasmids were co-transfected with the pCI-EGFP plasmid (ratio 3:1, respectively [16]). We found that there were no statistically significant differences between the cells expressing 3CL and m3CL concerning both fraction of EGFP-positive cells and mean fluorescence intensity of these cells (S1 Table). Additionally, mitochondrial metabolic activity and membrane integrity of EGFP-expressing cells were estimated by flow cytometry at 24 h and 48 h p.t. using

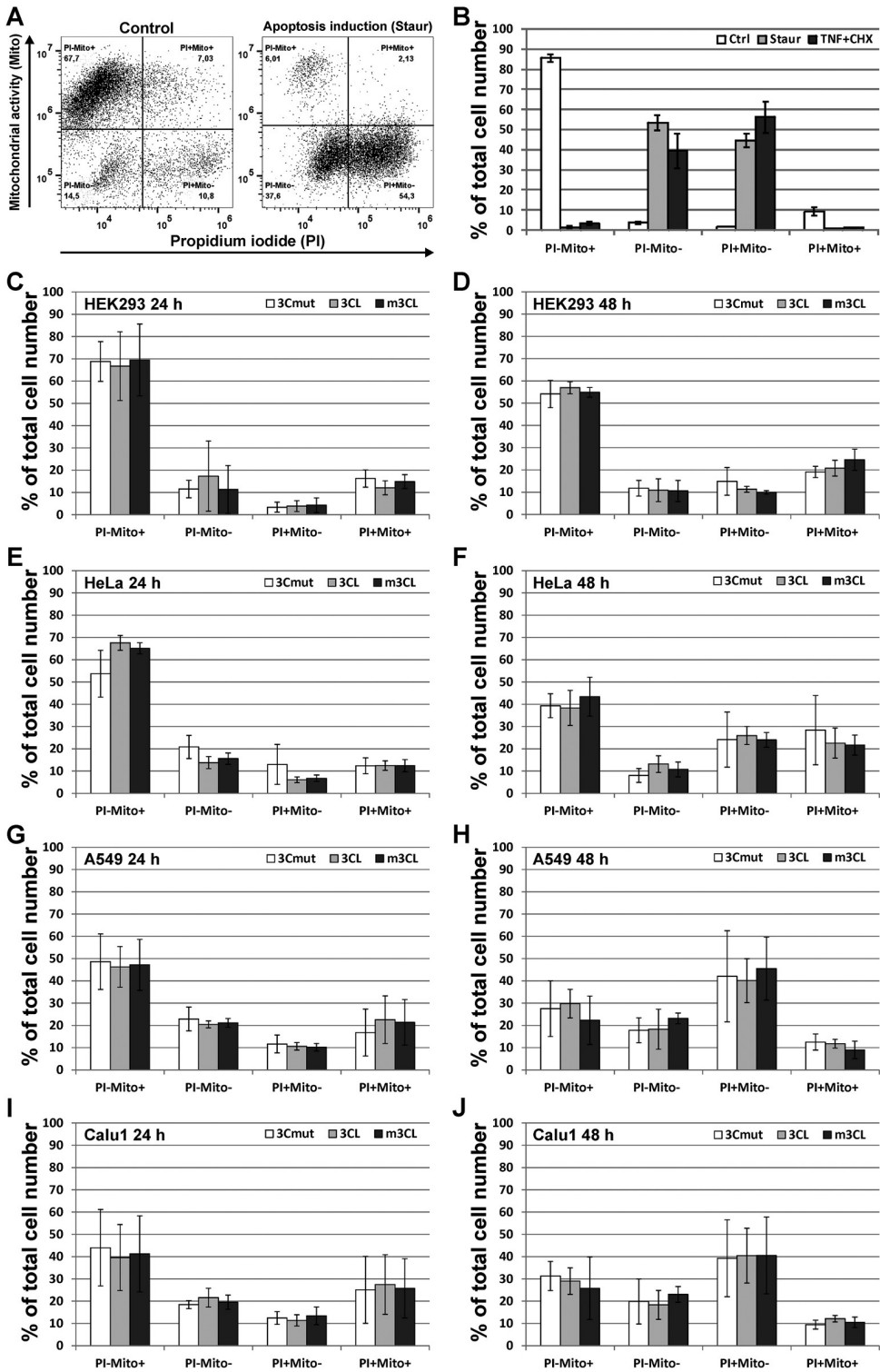

**Fig 3. Analysis of the cell viability of transfected cells using flow cytometry.** The mitochondrial metabolic activity and membrane integrity of the cells were estimated using 1,1′,3,3,3′,3′-hexamethylindodicarbocyanine iodide dye (Mito) and propidium iodide (PI), respectively. Representative dot plots (A) and representative results of the flow cytometry analysis (B) of the control non-transfected A549 cells and A549 cells treated with apoptosis inductors staurosporine (Staur) or tumor necrosis factor α in combination with cycloheximide (TNF+CHX). The cells were transfected with the p3Cmut (3Cmut), p3CL (3CL), and pm3CL (m3CL) plasmids mixed with the pCI-EGFP plasmid (ratio 3:1, respectively) and EGFP-expressing cells were analyzed using Mito and PI at 24 h (C, E, G, I) and 48 h (D, F, H, J) post transfection. Values are represented as mean ± SD of two independent experiments with triplicates (n = 6).

1,1′,3,3,3′,3′-hexamethylindodicarbocyanine iodide dye (Mito) and propidium iodide (PI), respectively (Fig 3A). In control non-treated cultures of all cell lines used the vast majority of the cells were characterized by metabolically active mitochondria and intact membrane (Mito-positive and PI-negative pattern) thus demonstrating normal viable state (Fig 3B). In the positive control cultures after 24 h of apoptosis stimulation viable cells were practically absent, while most of the cells were represented by two populations: early apoptotic cells having inactive mitochondria but intact membrane (Mito-negative and PI-negative pattern) and late apoptotic cells with compromised membrane (Mito-negative and PI-positive pattern) (Fig 3B). In case of the cultures transfected with p3CL, pm3CL and p3Cmut the predominant cell population at 24 h p.t. was presented by normal viable cells (Fig 3C, 3E, 3G and 3I). Noteworthy that compared to 24 h p.t. at 48 h p.t. there was a significant decrease in the fraction of the viable cells and proportional increase in fractions of the early and late apoptotic cells (Fig 3D, 3F, 3H and 3J). However, this decrease was not associated with the protein expressed and was observed for all plasmids thus likely reflecting the cytotoxic effect of the transfection and/or of the heterologous protein overexpression.

In summary, the results of the present study indicate that SARS-CoV-2 3CL$^{pro}$ doesn't induce cell death upon individual expression in human cells *in vitro*. Considering that SARS-CoV and SARS-CoV-2 share 96% sequence identity in 3CL$^{pro}$, the rest 4% may be responsible for the difference in the cytotoxicity between these two proteases. Although 3CL$^{pro}$ from SARS-CoV was not used is the current study, it was previously shown to induce apoptosis. Therefore, it is possible that the reason originates from the different experimental systems used, e.g., Vero-E6 cells are more sensitive to the 3CL$^{pro}$ action than the cell lines used in the present study. However, whatever the true reason for the differences found, in the present study we demonstrated that SARS-CoV-2 3CL$^{pro}$ is unlikely directly contribute to the cytopathic effect observed during viral infection.

## Supporting information

**S1 Fig. Specificity of primer/probe sets used for PCR analysis of the expression of 3CLpro/ m3CLpro (A) and UBC (B).** HEK293, HeLa, A549 and Calu1 cells were transfected with p3CL and pm3CL plasmids, and total RNA was isolated 24 h post transfection. The expression of 3CLpro, m3CLpro and ubiquitin C (UBC) genes was detected using quantitative real-time PCR with reverse transcription, and specificity of primer/probe sets was confirmed using corresponding melting curves.
(PDF)

**S2 Fig. 3CLpro proteolytic activity analysis.** Cells were co-transfected with p3CL, pm3CL, or pCI constructs and pGlo-3CL providing the expression of 3CLpro-specific biosensor; 24 h post transfection luciferase activity in transfected cultures was analyzed using GloSensor reagent. Values are represented as mean ± SD of two independent experiments with triplicates (n = 6).
(PDF)

**S1 Table. Characteristics of the cell cultures co-transfected with pCI-EGFP and p3CL/ pm3CL plasmids.** Values are represented as mean ± SD of two independent experiments with triplicates. No statistically significant differences between the cells expressing 3CL and m3CL were observed (nonparametric two-tailed Mann-Whitney U test, n = 6). h p.t.–hours post transfection.
(PDF)

## Acknowledgments

The work was carried out using the equipment of the Center of Cellular and Gene Technology of the Institute of Molecular Genetics of the National Research Centre "Kurchatov Institute".

## Author Contributions

**Conceptualization:** Alexey Komissarov, Ilya Demidyuk.

**Formal analysis:** Alexey Komissarov.

**Funding acquisition:** Sergey Kostrov, Ilya Demidyuk.

**Investigation:** Alexey Komissarov, Maria Karaseva, Marina Roschina.

**Methodology:** Alexey Komissarov.

**Project administration:** Ilya Demidyuk.

**Supervision:** Sergey Kostrov, Ilya Demidyuk.

**Writing – original draft:** Alexey Komissarov.

**Writing – review & editing:** Alexey Komissarov, Sergey Kostrov, Ilya Demidyuk.

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
