## [Decision Letter · Decision Letter 0]

23 Dec 2021

PONE-D-21-20749The SARS-CoV-2 main protease doesn’t induce cell death in human cells in vitroPLOS ONE

Dear Dr. Komissarov,

Thank you for submitting your manuscript to PLOS ONE. After careful consideration, we feel that it has merit but does not fully meet PLOS ONE’s publication criteria as it currently stands. Therefore, we invite you to submit a revised version of the manuscript that addresses the points raised during the review process. Please address the reviewer's concerns below

We look forward to receiving your revised manuscript.

Kind regards,

Irina V. Lebedeva, Ph.D.

Academic Editor

PLOS ONE

Journal Requirements:

Reviewers' comments:

Reviewer's Responses to Questions

**Comments to the Author**

1. Is the manuscript technically sound, and do the data support the conclusions?

Reviewer #1: Yes

2. Has the statistical analysis been performed appropriately and rigorously? 

Reviewer #1: Yes

3. Have the authors made all data underlying the findings in their manuscript fully available?

Reviewer #1: Yes

4. Is the manuscript presented in an intelligible fashion and written in standard English?

Reviewer #1: Yes

5. Review Comments to the Author

Reviewer #1: This manuscript examined the cell cytotoxicity induced by the SARS-CoV-2 main protease. It was found that the SARS-CoV-2 Mpro does not induce cytotoxicity in the four cell lines tested including HEK293, HeLa, A549 and Calu1. It was claimed that SARS-CoV-2 Mpro might be different from SARS-CoV Mpro in inducing apoptosis. The comments are:

1) No SARS-CoV Mpro was included as a control in this experiment, so the authors should be cautious in claiming that SARS-CoV-2 Mpro might be different from SARS-CoV Mpro.

2) SARS-CoV-2 Mpro has been previously expressed in cells in the Flip-GFP assay, and no cytotoxicity was observed. The cellular activity of Mpro was supported by the GFP signal. The following references should be cited:Xia Z, Sacco M, Hu Y, Ma C, Meng X, Zhang F, Szeto T, Xiang Y, Chen Y, Wang J. Rational Design of Hybrid SARS-CoV-2 Main Protease Inhibitors Guided by the Superimposed Cocrystal Structures with the Peptidomimetic Inhibitors GC-376, Telaprevir, and Boceprevir. ACS Pharmacol Transl Sci. 2021 Jun 9;4(4):1408-1421. doi: 10.1021/acsptsci.1c00099. PMID: 34414360; PMCID: PMC8204911.

Li X, Lidsky P, Xiao Y, Wu CT, GarciaKnight M, Yang J, Nakayama T, Nayak JV, Jackson PK, Andino R, Shu X. Ethacridine inhibits SARS-CoV-2 by inactivating viral particles in cellular models. bioRxiv [Preprint]. 2020 Nov 2:2020.10.28.359042. doi: 10.1101/2020.10.28.359042. PMID: 33140048; PMCID: PMC7605555.

3) It should be noted that another study showed that expression of Mpro with the GFP reporter induced cell death. In this study 293T cell was used, which is similar to the cell line used in the author's study. The author should provide a plausible explanation.

https://www.biorxiv.org/content/10.1101/2021.06.08.447613v1

4) The Protease-Glo luciferase assay reported in this study is novel. It should be pointed out that similar study was recently reported independently by another group. It should be cited.

https://www.biorxiv.org/content/10.1101/2021.08.28.458041v1

6. PLOS authors have the option to publish the peer review history of their article (what does this mean?). If published, this will include your full peer review and any attached files.

Reviewer #1: No

---

## [Author Response · Author response to Decision Letter 0]

28 Dec 2021

Together with co-authors we would like to thank the Reviewer for thorough examination of our work. Point-to-point response is presented below.

Response to the Reviewer #1 comments:

– “1) No SARS-CoV Mpro was included as a control in this experiment, so the authors should be cautious in claiming that SARS-CoV-2 Mpro might be different from SARS-CoV Mpro.”

In accordance with the Reviewer’s comment, we have discussed the comparison of the cytotoxic effects of SARS-CoV vs. SARS-CoV-2 3CL proteases more carefully (page 12, lines 269-276).

– “2) SARS-CoV-2 Mpro has been previously expressed in cells in the Flip-GFP assay, and no cytotoxicity was observed. The cellular activity of Mpro was supported by the GFP signal. The following references should be cited:Xia Z, Sacco M, Hu Y, Ma C, Meng X, Zhang F, Szeto T, Xiang Y, Chen Y, Wang J. Rational Design of Hybrid SARS-CoV-2 Main Protease Inhibitors Guided by the Superimposed Cocrystal Structures with the Peptidomimetic Inhibitors GC-376, Telaprevir, and Boceprevir. ACS Pharmacol Transl Sci. 2021 Jun 9;4(4):1408-1421. doi: 10.1021/acsptsci.1c00099. PMID: 34414360; PMCID: PMC8204911.

Li X, Lidsky P, Xiao Y, Wu CT, GarciaKnight M, Yang J, Nakayama T, Nayak JV, Jackson PK, Andino R, Shu X. Ethacridine inhibits SARS-CoV-2 by inactivating viral particles in cellular models. bioRxiv [Preprint]. 2020 Nov 2:2020.10.28.359042. doi: 10.1101/2020.10.28.359042. PMID: 33140048; PMCID: PMC7605555.”

We have added the discussion of the controversy of the data concerning the cytotoxicity of SARS-CoV-2 3CLpro and cited corresponding references (page 3, lines 44-50).

– “3) It should be noted that another study showed that expression of Mpro with the GFP reporter induced cell death. In this study 293T cell was used, which is similar to the cell line used in the author's study. The author should provide a plausible explanation.

https://www.biorxiv.org/content/10.1101/2021.06.08.447613v1”

Unfortunately, it is very hard to interpret properly the results of this preprint. In this paper the authors used GFP fluorescence intensity as a marker of cell viability, which is incorrect. Results presented on Figure 2D indicate that the higher the concentration of the MPI8 (3CLpro inhibitor), the higher the GFP fluorescence of transfected cells, and these data are interpreted by the authors as evidence of dose dependent effect between 3CLpro activity and its cytotoxicity. Unfortunately, it is incorrect, since it can be seen on the Figure 2C that MPI8 treatment results in the increase in GFP signal without affecting the fraction of GFP expressing cells (that is approximately 20% even in the absence of MPI8). These results indicate that 3CLpro activity affects GFP expression level and/or GFP fluorophore maturation kinetics. Since cell morphology and key markers of cell viability are not analyzed, no conclusion can be made concerning of the live/dead status of these cells. Besides, on the Figure 2B we can see morphology of the transfected cells without MPI8 treatment, and they demonstrate no signs of death like loss of adhesion, rounding, floating and etc. Therefore, we believe that the results of the current preprint don’t provide the data about the presence of cytotoxicity of 3CLpro.

- “4) The Protease-Glo luciferase assay reported in this study is novel. It should be pointed out that similar study was recently reported independently by another group. It should be cited.

https://www.biorxiv.org/content/10.1101/2021.08.28.458041v1.

Manuscript text has been edited in accordance with the Reviewer’s comment (page 9, lines 194-195).

---

## [Editor Report · Decision Letter 1]

14 Mar 2022

The SARS-CoV-2 main protease doesn’t induce cell death in human cells in vitro

PONE-D-21-20749R1

Dear Dr. Komissarov,

We’re pleased to inform you that your manuscript has been judged scientifically suitable for publication and will be formally accepted for publication once it meets all outstanding technical requirements.

Kind regards,

Irina V. Lebedeva, Ph.D.

Academic Editor

PLOS ONE
---

## [Editor Report · Acceptance letter]

16 May 2022

PONE-D-21-20749R1 

The SARS-CoV-2 main protease doesn’t induce cell death in human cells *in vitro*

Dear Dr. Komissarov:

I'm pleased to inform you that your manuscript has been deemed suitable for publication in PLOS ONE. Congratulations! Your manuscript is now with our production department. 

Kind regards, 

on behalf of

Dr. Irina V. Lebedeva 

Academic Editor

PLOS ONE